# MultiHot Embedding: A Multiple Activation Embedding Model for Numerical Features in Deep Learning

## Abstract

Numerical feature learning has long been a challenging problem in deep learning. Deep learning models exhibit sub-optimal performance in many numerical-feature-intense learning tasks. This paper proposes a simple but effective method, i.e. MultiHot Embedding, for numerical feature representation in deep learning models. The MultiHot Embedding discretizes the numerical data into bins and extends the one-hot embedding by allowing multiple activations of neighbor bits. The multiple neighbor activation mechanism enables the MultiHot Embedding to use small bin widths for discretization which overcomes the information loss problem as well as the inadequate training issue. The experiments on 6 numerical feature learning tasks validate the effectiveness and generalization capabilities of the proposed MultiHot Embedding method. Compared to the baseline models, the MultiHot Embedding model significantly improves the prediction performance. Specifically, it outperforms the state-of-the-art numerical feature representation model which has a much more complex structure. Furthermore, the sensitivity analysis shows that the MultiHot Embedding is capable of handling small width discretization width, which effectively reduces the information loss during the discretization process.

## 1 Introduction

Numerical feature learning has posed a persistent challenge within the realm of deep learning Gorishniy et al. (2022); Ke et al. (2019); Guo et al. (2021). Extensive studies reported that deep learning models exhibit sub-optimal performance in many numerical-feature-intense tasks, such as Click-Through Rate (CTR) prediction Ke et al. (2019); Guo et al. (2021), tabular data learning Gorishniy et al. (2021), etc., where the input contains plenty of numerical fields. For example, tree-based methods such as Gradient Boosting Decision Trees (GBDT) still dominate most of the tabular data learning tasks Grinsztajn et al. (2022), which even beat the state-of-the-art (SOTA) deep learning models. On the other hand, deep learning models are expertise in categorical feature learning. For example, Natural Language Processing (NLP) tasks are dominated by deep learning models, which outperform other methods by a significant margin. The root of the excellent performance is the embedding of categorical features. It is widely proved that embedding, i.e. transforming the input feature into a dense semantic vector through a look-up table at the beginning of the network, is of great benefit for categorical feature learning. It has been adopted as a standard procedure for plenty of NLP models Vaswani et al. (2017); Devlin et al. (2018). Based on these facts, it is natural to explore whether categorizing and embedding numerical features could similarly enhance the performance.

Following the pipeline of categorical feature embedding, numerical inputs can be embedded after discretization, which we refer to as Discretization and Embedding (D&E). Compared to directly using the numerical value[1], D&E exhibits great potential. Firstly, discretization confines input numerical values to enumerable bins, thereby reducing learning complexity and enhancing robustness against outliers. Secondly, each bin can be treated as a category, and through embedding, trans-

---

[1]In this paper, utilizing the original numerical values also encompasses cases where values undergo initial preprocessing, e.g., normalization or standardization.

formed into a semantic vector; this embedding conveys more information compared to the original numerical value. Thirdly, the embedding-based approach can be seamlessly integrated into various deep learning-based numerical feature learning models without altering the core architecture. However, in contrast to categorical feature embedding that retains all information, discretization of numerical value incurs information loss, i.e. values within the same bin are treated uniformly, resulting in the loss of numerical distinctions. As a result, to maximize the benefits of numerical feature embedding, it is important to mitigate the loss of numerical information during the D&E process.

Minimizing information loss in D&E poses a great challenge. Achieving a more profound reduction in information loss necessitates the adoption of narrower bin widths. A reduced bin width signifies diminished numerical variance within each bin, thereby mitigating the impact of information loss. Nevertheless, opting for smaller bins concurrently escalates the bin count, subsequently impeding the adequate training of the associated embedding due to the limited training samples distributed in each bin (elaborated in Section 2). Consequently, prevailing D&E methods encounter constraints in pursuing narrower bin widths to attain enhanced performance upgrades.

In this paper, we address this gap by introducing the MultiHot Embedding, a simple but effective D&E approach that enables narrow discretization in the numerical feature embedding. MultiHot Embedding employs the notion of "overlap" to depict the numeric differentiation (a facet absent in traditional discretization methods). Furthermore, the method significantly enhances training effectiveness, particularly with a narrow bin width, through the utilization of the overlapped MultiHot activation approach. The main contribution of this study is summarized as follows:

- Dissect the pros and cons of discretization in the numerical feature learning.
- Propose a MultiHot Embedding method for the numerical value representation in deep learning, which overcomes the inadequate training issue in D&E of numerical input.
- Evaluate the proposed method performance on 6 numerical feature learning tasks and conduct the sensitivity analysis on embedding parameters.

## 2 PROS AND CONS OF DISCRETIZATION

Discretization confines numerical values into bins. When embedding is conducted after discretization, values within a given bin share a common representation (embedding). Consequently, the model only needs to establish a representation for each bin without addressing numerical variations within them. This notably reduces the complexity of learning. Furthermore, outliers can be addressed through discretization (e.g. outlier 150000 still falls into the bin labeled as *larger than 100*). Nonetheless, the discretization of numerical values also presents certain drawbacks. The discretization process theoretically gives rise to two critical concerns: *SBD* (Similar value But Dis-similar discretization) and *DBS* (Dis-similar value But Same discretization) Guo et al. (2021):

- *SBD*. Discretization may separate similar values (boundary values) into different bins. For example, a common discretization of the *Time of Day* is dividing a day into 24 one-hour bins (e.g. 8:00-8:59 am is discretized as 8) Feng et al. (2022). In this case, 8:59 am and 9:00 am will be discretized into two different bins (i.e. 8 and 9) although they are close and share similar context information (e.g., peak hour).
- *DBS*. Discretization may group significantly different values into the same bin. Using the same example as above, 8:00 am and 8:59 am are in the same bin, although they represent different context information. That is, 8:00 am is close to the off-peak hour and 8:59 am is in the morning peak period.

Discretizing the numerical feature can be optimally beneficial if the drawbacks (namely, *SBD* and *DBS* issues) are appropriately addressed. For the discretization process, the most effective way of reducing the *SBD* and *DBS* problem is adopting a narrower bin width. When the bin width is sufficiently small, the values within a bin tend to exhibit greater similarity, thereby enhancing the consistency of patterns within each bin. For instance, consider a 5-minute bin employed in the time-of-day discretization. Even though 8:59 am and 9:00 am still fall into distinct bins (i.e. 8:55-8:59 and 9:00-9:05, respectively), the patterns of these two bins resemble each other more closely than in the scenario of 1-hour width bins (i.e. 8:00-8:59 and 9:00-9:59). This observation points to a considerably weaker influence of the *SBD* concern.

However, there is no free lunch. A small bin width gives rise to an additional problem: inadequate training. The reduced width implies a larger quantity of bins, resulting in a decreased number of training samples distributed within each bin. Consequently, the model's training becomes inadequate to effectively embed the characteristics of the bins, as shown in Figure 1.

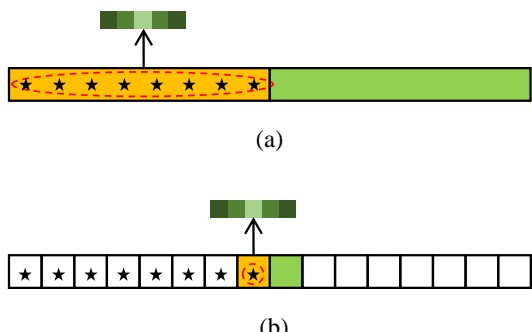

(a)

(b)

Figure 1: Example of inadequate training problem. When a small bin width is adopted (Figure 1 (b)), training samples (black stars) are limited compared to the wider bin width scenario (Figure 1 (a)), which may result in the poor semantic quality of the learned embedding vector.

By far, we summarize the issues for D&E as follows. 1) The input value discretization reduces the learning complexity but causes *SBD* and *DBS* problems. 2) Using small bin width mitigates the *SBD* and *DBS* problems, but causes inadequate training towards the representation (embedding) of bins.

## 3    MULTIHOT EMBEDDING

In this section, we propose a MultiHot Embedding method that addresses the issues of the *SBD* and *DBS* from the value discretization, as well as the inadequate training problem when using a small bin width.

### 3.1    METHODOLOGY

As discussed, using a small discretizing bin width could reduce the *SBD* and *DBS* issues. Therefore, the MultiHot Embedding utilizes a relatively small bin width compared to the conventional discretization method. For example, in the case of the *Time of Day* discretization, the bin width could be set as 10 minutes (compared to the 1 hour bin in conventional methods). To clarify the illustration, we denote the discretization using a normal bin width $l_N$ as $D_N(\cdot)$, and $D_S(\cdot)$ to represent the one with a small bin width $l_S \ll l_N$. After discretization, the range of feature $C$ is divided into $K$ bins $\{I_1, I_2, \ldots, I_K\}$. For bin $I_i \in \{I_1, I_2, \ldots, I_K\}$, we denote its range as $[c_i, c_{i+1}]$, where $c_i$ and $c_{i+1}$ are the upper and lower bounds of $I_i$, respectively. Note that the bins are arranged in an ordinal sequence, i.e. $c_1 < c_2 < \cdots < c_K$. For a specific numerical value $c$ within an bin $I_i$, we denote its one-hot representation under $D_S(\cdot)$ as:

$$\mathbf{c}_o^S = onehot(D_S(c)) = [o_1, o_2, \ldots, o_K] \tag{1}$$

$$o_t = \begin{cases} 1, & t = i \\ 0, & \text{otherwise} \end{cases} \tag{2}$$

$$\mathbf{E}_{\mathbf{c}_o^S} = \mathbf{c}_o^S \mathbf{M} \tag{3}$$

where $\mathbf{c}_o^S \in R^{1 \times K}$ is the one-hot representation of $c$ under $D_S(\cdot)$, $o_t$ the $t^{th}$ ($t \in [1, K]$) element of $\mathbf{c}_o^S$, and $i$ the index of the discretization bin. $\mathbf{E}_{\mathbf{c}_o^S} \in R^{1 \times h}$ is the corresponding embedding of $\mathbf{c}_o^S$, $\mathbf{M} \in R^{K \times h}$ the learnable look-up matrix, and $h$ the embedding size.

As we use a small bin width $l_S$, $\mathbf{E}_{\mathbf{c}_o^S}$ is sub-optimal due to the inadequate training issue. To address that, we propose the MultiHot representation of $c$ as:

$$\mathbf{c}_m^S = MultiHot(D_S(c)) = [m_1, m_2, \ldots, m_K] \tag{4}$$

$$m_t = \begin{cases} 1, & max(i-m,1) \leq t \leq min(i+m,K) \\ 0, & \text{otherwise} \end{cases} \tag{5}$$

$$\mathbf{E}_{\mathbf{c}_m^S} = \mathbf{c}_m^S \mathbf{M} \tag{6}$$

where $m$ is a non-negative integer predefined specifying the degree of neighbourhood information sharing and $m_t$ is the $t^{th}$ ($t \in [1, K]$) element of $\mathbf{c}_m^S$. $\mathbf{E}_{\mathbf{c}_m^S} \in R^{1 \times h}$ is the MultiHot embedding of the numerical value $c$

The MultiHot representation is transformed from one-hot by additionally activating the $m$-neighbors of $i$ (i.e. set $m$-neighbors of $i$ as 1). For example, suppose an one-hot representation of $D_S(c)$ is $[0, 0, 0, 0, 1, 0, 0, 0]$ where $i = 5$, then the corresponding MultiHot representation is $[0, 0, 1, 1, 1, 1, 1, 0]$ with $m = 2$. Note that when $m = 0$, the MultiHot representation is identical to the one-hot.

## 3.2 PROPERTIES

We discuss the MultiHot Embedding properties and illustrate them using an example.

**Property 1**: *The MultiHot Embedding retains the numerical relationship under the discrete context leveraging the "overlapping" concept, which addresses the root causes of SBD and DBS issues in discretizing and representing numerical features.*

Using the discretization of the *Time of Day* as an example, as discussed before, the embeddings of 8:59 am and 9:00 am are different under a one-hour discretization, which raises the SBD issue. The MultiHot Embedding uses a small discretization bin width (e.g. 10 minutes) and the $m$-neighbor activation mechanism. Therefore, most of the "1" in their MultiHot representations are overlapped, resulting in similar embeddings (Figure 2), which addresses the *SBD* issue. Following the same analysis logic, given the small discretization length, the MultiHot Embeddings of 8:00 am and 8:59 am are rather different as what they are supposed to be in representing different context information. Therefore, the *DBS* problem is overcome to a certain level. In all, the MultiHot Embedding captures numerical relationships between numerical values in representing their context difference by overlapping their one-hot representations to $m$-neighbors. The traditional discretization & embedding models basically lose the numerical feature of numerical variable values.

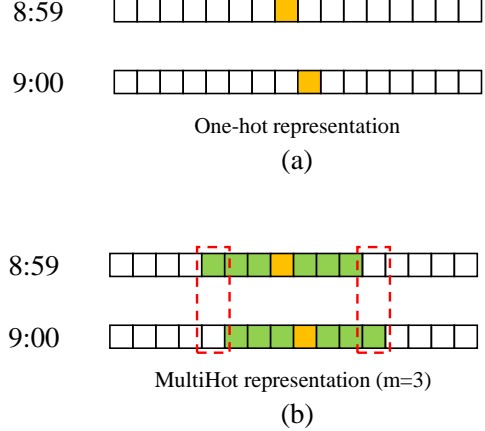

Figure 2: One-hot and MultiHot representations for 8:59 and 9:00, respectively. One-hot representations of 8:59 and 9:00 are totally different although they are very similar in the context (Figure 2 (a)), while in MultiHot, most of the activated elements (colored boxes) are overlapped, resulting in similar context values after embedding.

**Property 2**: *The MultiHot Embedding leads to more efficient training than the one-hot, thus allowing for using a small discretization bin width.*

As mentioned, the *SBD* and *DBS* problems can be reduced by using a small bin width in discretization. However, conventional discretization approaches cannot handle small bin widths due to the consequent inadequate training issue. For example, if the 10-minutes bin width is used in the *Time of Day* discretization, the one-hot embedding of a large number of bins may not be well-trained due to inadequate training sample sizes. In the MultiHot Embedding representation, the training of a specific discretization bin will activate the bit itself and its $m$ neighbors, thus the embedding of each discretization bin gets more chance to be trained (explicitly $2m$ times more than the one-hot). In other words, the MultiHot Embedding functions are equivalent to increasing the number of training samples for each discretization bin.

In summary, the MultiHot Embedding model would conceptually overcome the main concerns of representation of numerical features in deep learning, including *SBD* and *DBS* problems from numerical data discretization, and inadequate training with small discretization bin width. We will further validate these in the case study using three typical types of learning tasks.

## 3.3 INTERPRETATION

In the one-hot representation, each element of the vector can be explained as whether the value belongs to the corresponding bin (1 for belonging and 0 otherwise). To better illustrate the MultiHot representation, we also provide a corresponding semantic explanation. Let $D_S(C) = \{[c_1, c_2], [c_2, c_3], \ldots, [c_{K-1}, c_K]\}$ be the discretization of numerical feature $C$. The $t^{th}$ element in $\mathbf{c}_m^S$ represents whether the value $c$ belongs to bin $[c_{t-m}, c_{t+1+m}]$ (1 for belonging and 0 otherwise). For comparison, we also give the corresponding definition of one-hot representation: the $t^{th}$ element in $\mathbf{c}_o^S$ represents whether the value $c$ belongs to bin $[c_t, c_{t+1}]$. Figure 3 shows the difference between the two representations. From this perspective, the main difference between one-hot and MultiHot representations is that the adjacent bins in one-hot are disjoint, while in MultiHot, they are overlapped.

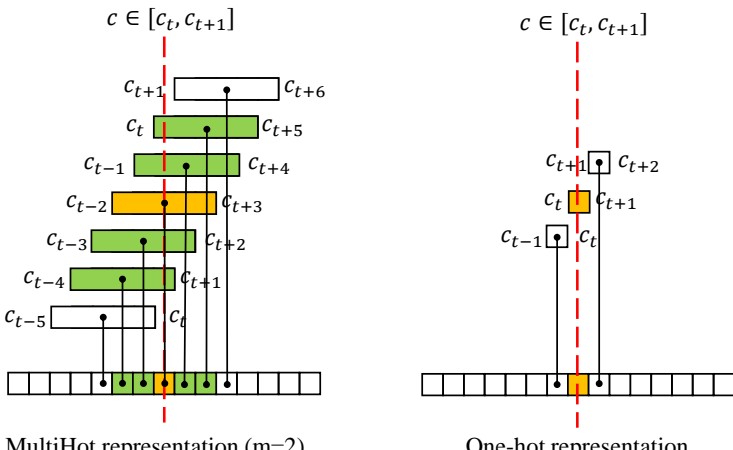

MultiHot representation (m=2)    One-hot representation

Figure 3: Explanation of MultiHot and one-hot representations. The values specify the upper and lower bounds of each bin. The black lines connect the vector elements and the corresponding bin they represent. The colored bins contains the given value $c$, of which their corresponding values are set as 1 in the vector. For one-hot, as the bins are non-overlapped, only one element is 1 in the vector.

## 4 EMPIRICAL EVALUATION

### 4.1 EXPERIMENTAL SETUP

We use a diverse set of 6 public datasets (specified in Gorishniy et al. (2021)) containing massive numerical input features. The datasets include: **California Housing** Pace & Barry (1997) (**CA**, real estate data), **Adult** Kohavi et al. (1996) (**AT**, income estimation), **Helena** Guyon et al. (2019) (**HE**, anonymized dataset), **Jannis** Guyon et al. (2019) (**JA**, anonymized dataset), **Higgs** Vanschoren et al. (2014) (**HI**, simulated physical particles), **Year** Bertin-Mahieux et al. (2011) (**YE**, audio features). We include the detailed dataset description in the supplementary.

We employ two neural network architectures, i.e. **Multi-Layer Perceptron** (MLP) and **ResNet** He et al. (2015), as the foundational models for the evaluation. MLP serves as the basic model for numerical feature learning tasks, while ResNet, originally developed for computer vision applications, has recently demonstrated remarkable prowess in numerical feature learning Gorishniy et al. (2021). Notably, we do not include more complicated models, such as transformer-based models, due to their sensitivity to hyperparameters and training settings. The performance fluctuation caused by such settings could potentially obscure the influence of different numerical feature representation methods.

- **Standardization** (STD). Making the values of numerical feature $C$ have zero-mean and unit-variance:

$$c' = \frac{c - \overline{c}}{\sigma} \tag{7}$$

  where $\overline{c}$ and $\sigma$ denote the mean and variance of feature $C$, respectively.

- **One-hot & Embedding** (OE). The range of the numerical feature $C$ is divided into $K$ bins with the same number of samples (i.e. equal frequency discretization). The values are then transformed into the corresponding embedding vectors.

- **AutoDis** (AD) Guo et al. (2021). **State-of-the-art** numerical feature representation method which combines a Meta-Embedding and Automatic Discretization mechanisms.

The model structure is shown in Figure 4. We follow the model configuration and training details specified in Gorishniy et al. (2021), which provides open-source well-trained models for these datasets as baselines[2].

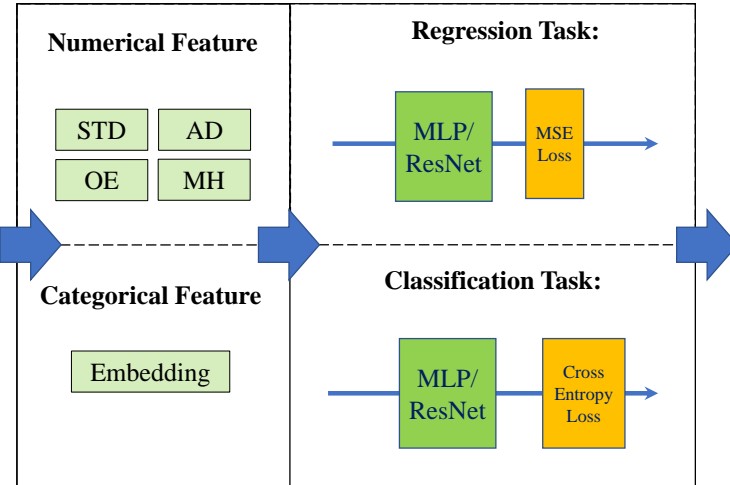

Figure 4: Model structure

---

[2]The evaluation experiments are open-soured at: https://github.com/qzl408011458/MultiHot-Embedding-in-Tabular-Learning

Table 1: Evaluation results. For each dataset, Best results (for MLP and ResNet respectively) are in bold. Notation: ↓ RMSE (the smaller the better), ↑ Accuracy (the larger the better), Improvement (%): performance improvement of MultiHot Embedding compared to the best baseline, the negative value indicates worse than the best baseline.

|  | CA ↓ | AT ↑ | HE ↑ | JA ↑ | HI ↑ | YE ↓ |
|---|---|---|---|---|---|---|
| MLP+STD | 0.499 | 0.852 | 0.383 | **0.719** | 0.723 | 8.853 |
| MLP+OE | 0.495 | 0.861 | 0.346 | 0.698 | 0.714 | 8.913 |
| MLP+AD | 0.488 | 0.855 | 0.385 | 0.716 | 0.723 | 8.793 |
| MLP+MH | **0.461** | **0.863** | **0.386** | 0.717 | **0.724** | **8.781** |
| Improvement (%) | 7.67 | 1.25 | 0.79 | -0.17 | 0.18 | 0.81 |
| ResNet+STD | 0.486 | 0.854 | 0.396 | **0.728** | 0.727 | 8.846 |
| ResNet+OE | 0.488 | 0.862 | 0.354 | 0.695 | 0.708 | 9.054 |
| ResNet+AD | 0.461 | 0.859 | 0.399 | 0.721 | **0.728** | 8.866 |
| ResNet+MH | **0.441** | **0.871** | **0.405** | 0.722 | **0.728** | **8.833** |
| Improvement (%) | 9.22 | 2.04 | 2.27 | -0.80 | 0.00 | 0.14 |

## 4.2 OVERALL PERFORMANCE

Table 1 presents the performance comparison results for various representation methods. The proposed MultiHot Embedding consistently outperforms the baselines in almost all tested datasets, achieving the best performance in five out of six datasets for both MLP and ResNet scenarios. This observation underscores the strong generalization capability and robustness of the proposed numerical feature representation method.

Specifically, it results in a performance increase of over 7% and 9% in the case of the CA dataset for MLP and ResNet, respectively. Furthermore, it leads to an improvement of over 2% for the AT and HE datasets when utilizing ResNet, which is notably significant.

Compared to the proposed MultiHot embedding model, AD has a much more complicated model structure but exhibits weaker performance, i.e. MultiHot embedding consistently improves the performance of all the evaluated datasets. The main reason is that AD still uses the numerical value as input without the discretization process. The learning complexity of numerical features limits its performance (as discussed in Section 2).

Among those baselines, OE exhibits a notable performance degradation compared to the MultiHot Embedding, although both of them fall within the D&E category. This indicates the significant side effect if the *SBD* and *DBS* problems are not well-addressed, i.e. the substantial information loss during discretization outweighs the benefit gained from reducing the complexity of learning.

## 4.3 SENSITIVITY ANALYSIS

We evaluate the impact of different discretization configurations on the model performance. Figure 5 summarizes the corresponding model performance of CA dataset across varying numbers of discretization bins.

The performance of OE firstly increases and then decreases as the number of bins gradually increases in both MLP and ResNet scenarios. This trend suggests that they are not capable of handling small bin width discretization. Their performance is sub-optimal with either a small or large number of discretization bins. The information loss issue is serious with few bins as it leads to *SBD* and *DBS* problems, while too many bins lead to the inadequate training issue (as shown in Figure 1), both resulting in unsatisfied performance. The results further reveal that the performance of the traditional D&E methods are more likely to fluctuate due to their sensitivity to parameter settings (i.e. the bin count determination). In other words, finding an optimal setting of the traditional D&E methods is exhausting. The performance of MultiHot Embedding achieve the best performance[3] at a

---

[3]The best performance here are worse than the performance specified in Table 1 since they are only the best among the evaluated bin counts in this experiment (i.e. 20, 40, 60, etc.).

relatively large bin count. This highlights the advantage of MultiHot Embedding in mitigating *SBD* and *DBS* problems, given its ability to effectively handle a small bin width in discretization.

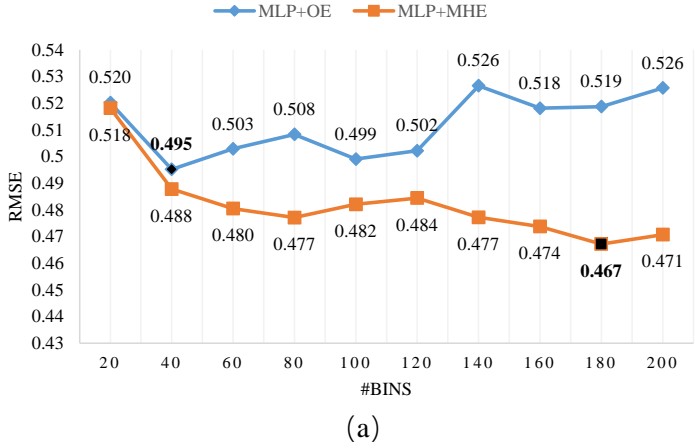

(a)

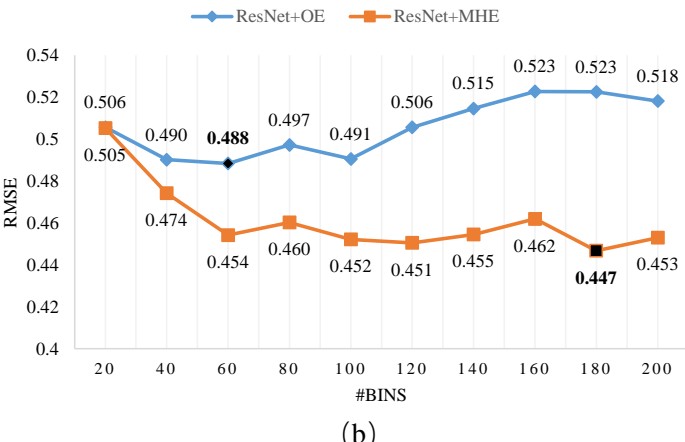

(b)

Figure 5: Sensitivity analysis of the number of discretization bins for MLP (Figure 5 (a)) and ResNet (Figure 5 (b)), respectively. The best result of each curve is in bold. The #BINS denotes the number of bins used in the model. OE achieves the best performance with a small number of bins and its performance degrades due to the lack of enough training samples at each bin (i.e. inadequate training issue). For the MultiHot Embedding, its performance consistently improves and achieves the best performance at a relatively large number of discretization bins.

## 5 RELATED WORK

### 5.1 FEATURE REPRESENTATION

The essence of deep learning models is automatically learning latent representations of input features. Representation learning is important since it can extract general priors about the physical world and data generation process, i.e., priors that are not task-specific but would be likely to be useful Bengio et al. (2013). For example, CNN Krizhevsky et al. (2012), the most widely used deep learning model in image processing, learns gradually the abstracted image representations layer by layer. Explicitly, the basic features, such as colors, edges, etc., are learned by shallow layers and synthesized to more specific representations as layers go deeply Krizhevsky et al. (2012). For the NLP models, such as Word2vec Mikolov et al. (2013), ELMo Peters et al. (2018), Transformer

Vaswani et al. (2017), endeavor to learn distributed representations of words and sentences in the semantic space, in which similar semantic meaning exhibits a close distance. However, limited attention has been paid to how to develop good representation structures for numerical features that involve proper and effective prior that facilitates the learning process.

## 5.2 DISCRETIZATION

Discretization is a commonly used preprocessing method for numerical input features Garcia et al. (2013). Many discretization methods were proposed in the literature, including the equal width and frequency discretization schemes, and approaches based on theories such as information entropy Dougherty et al. (1995), statistical $\mathcal{X}^2$ test Kerber (1992); Liu & Setiono (1997), likelihood Wu (1996); Boullé (2006), rough set Nguyen & Skowron (1995); Zhang et al. (2004), etc. However, these approaches suffer from the *SBD* and *DBS* problems since they cannot handle small bin widths due to the inadequate training issue. It is worth mentioning that the fuzzy discretization Shanmugapriya et al. (2017) also considers a value's membership of multiple bins. However, the fuzzy logic only fits the dataset with a vagueness nature, thus it is not generic to a wide range of deep learning tasks.

## 6 CONCLUSION

In this paper, we propose a MultiHot Embedding model for the representation of numerical features in deep learning models. The MultiHot Embedding use a simple but effective mechanism to add a $m$-neighbour activation mechanism upon the OneHot representation. It addresses two main issues of numerical feature representations, including the *SBD & DBS* problems from discretization, and inadequate training with small discretization bin widths. The multiple activation mechanism enables the MultiHot Embedding to utilize relatively a small bin width during discretization which overcomes the inadequate training problem. The experiments on 6 numerical feature learning tasks validate the effectiveness and generalization capabilities of the proposed numerical feature representation model. Compared to the best baseline models, the MultiHot Embedding model significantly improves the prediction performance by more than 7% for CA dataset, and more than 2% for AT and HE. The sensitivity analysis highlights the ability to handle small bin width discretization of the proposed MultiHot Embedding model, which is crucial in addressing *SBD* and *DBS* problems in discretization.

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
