# SUPPLEMENTAL MATERIAL

## 1 SOFTWARE AND HARDWARE

All model-dataset pairs are trained and evaluated on the workstation installed with NVIDIA RTX A5000 24 Gb. All the experiments are conducted under the same virtual Python environment. More details can be found in the repository `https://github.com/qzl408011458/MultiHot-Embedding-in-Tabular-Learning`.

## 2 DATA

Table 1: Datasets description. # Train, # Validation, and # Test respectively denote how many samples are used in the three datasets. # Num and # Cat represent the numbers of numerical features and categorical features.

| Name | Abbr | # Train | # Validation | # Test | # Num | # Cat | Task type | Batch size |
|------|------|---------|--------------|--------|-------|-------|-----------|------------|
| California Housing | CA | 13209 | 3303 | 4128 | 8 | 0 | Regression | 256 |
| Adult | AT | 26048 | 6513 | 16281 | 6 | 8 | Binclass | 256 |
| Helena | HE | 41724 | 10432 | 13040 | 27 | 0 | Multiclass | 512 |
| Jannis | JA | 53588 | 13398 | 16747 | 54 | 0 | Multiclass | 512 |
| Higgs Small | HI | 62752 | 15688 | 19610 | 28 | 0 | Binclass | 512 |
| Year | YE | 370972 | 92743 | 51630 | 90 | 0 | Regression | 1024 |

## 3 HYPERPARAMETER SEARCHING

The structures and hyper-parameters of backbones MLP and ResNet are both referred to the best benchmarks of Gorishniy et al. (2021) (i.e., MLP+STD and ResNet+STD). We use grid searching to train models with multiple hyper-parameter combinations and select the best ones evaluated in testset. The parameter searching settings of feature representation methods are shown in the table 2 and the best hyper-parameters for the corresponding models are selected as presented in table 3.

Table 2: Grid searching settings. $h$, $K$ and $m$ respectively denote embedding size, discretized bins and neighbors that are mentioned in our formal paper. $\tau$ is the temperature coefficient which is a key hyper-parameter of the AD proposed in the Guo et al. (2021). Each triple in the table represents the lower bound, upper bound and searching step size.

| | $h$ | $K$ | $m$ | $\tau$ |
|--|-----|-----|-----|--------|
| MLP+OE | (2, 30, 2) | (10, 200, 2) | NA | NA |
| MLP+AD | (2, 30, 2) | (10, 200, 2) | NA | (0.01, 0.5, 0.01) |
| MLP+MH | (2, 30, 2) | (10, 200, 2) | (5, 40, 1) | NA |
| ResNet+OE | (2, 30, 2) | (10, 200, 2) | NA | NA |
| ResNet+AD | (2, 30, 2) | (10, 200, 2) | NA | (0.01, 0.5, 0.01) |
| ResNet+MH | (2, 30, 2) | (10, 200, 2) | (5, 40, 1) | NA |

Table 3: The best hyper-parameters of the models evaluated in the paper.

|  | $h$ | $K$ | $m$ | $\tau$ |
|---|---|---|---|---|
| MLP+OE | 20 | 40 | NA | NA |
| MLP+AD | 20 | 100 | NA | 0.05 |
| MLP+MH | 2 | 192 | 7 | NA |
| ResNet+OE | 20 | 60 | NA | NA |
| ResNet+AD | 20 | 100 | NA | 0.05 |
| ResNet+MH | 2 | 190 | 24 | NA |