# OpenReview forum: "MultiHot Embedding: A Multiple Activation Embedding Model for Numerical Features in Deep Learning"
_ICLR.cc/2024/Conference — Submitted to ICLR 2024_

### Official Review · Reviewer_4mSv · 2023-10-31

**Soundness:** 2 fair
**Presentation:** 2 fair
**Contribution:** 2 fair
**Rating:** 3
**Confidence:** 4

**Summary:**

The paper propose a method for embedding numerical features into overlapping discrete bins. The paper evaluates their method on some tasks.

**Strengths:**

The paper tackles an under-explored space which has several potential application.

**Weaknesses:**

The paper is very limited in scope. For example, I don't think that highlighting the "pros and cons" of discretization is a valid contribution for a venue such ICLR, althought it could serve as a motivation paragraph in the introduction.

While the method seems novel to me, the comparison to prior works is very limited. The paper does not compare against published number in the literature making it hard to judge whether the empirical improvement comes from the method itself or from the fact that the other methods are under-tuned. For example, it argues that AutoDis is the state-of-the art method, then why not comparing the proposed method against the benchmarks published in that paper?

The paper lacks details about the experimental setup. What are the hyperparameters that lead to the performance improvement? Are they the same across the benchmarks?

Finally, the paper should present an analysis of the effect of changing the different parameters of their method. For example, what happens when one changes $m$, the number of neighbors?

**Questions:**

* What are the hyperparameters that lead to the performance improvement? Are they the same across the benchmarks?
* What happens when one changes $m$, the number of neighbors?
* How does the method compares to the benchmarks used in prior work and their published numbers?

---

### Official Review · Reviewer_41SS · 2023-11-02

**Soundness:** 2 fair
**Presentation:** 2 fair
**Contribution:** 1 poor
**Rating:** 3
**Confidence:** 4

**Summary:**

The paper presents a multi-hot embedding method for numerical features in deep learning.

**Strengths:**

1. The proposed method is well-motivated.
2. The experiments showed that the proposed method is more effective than the baselines on a few datasets.

**Weaknesses:**

1. The problem studied in the paper is not interesting at all. In modern deep learning, numerical features can be well handled by neural networks. The necessity of embedding before learning is tiny.
2. The proposed method is trivial.
3. Sections 3.2 and 3.2 do not provide any concrete result such as the theoretical information loss of the proposed discretization method.
4. The numerical results are not sufficient. Please refer to my questions.

**Questions:**

1. In one-hot encoding, the sum of the embedding is a constant one. In multi-hot encoding, is the sum always a constant? If not, the embedding is not fair for different features.
2. What is the theoretical information loss of the proposed discretization method?
3. In Table 1, the meaning of the columns is not clear to me at all. Are they datasets? In the caption, RMSE and Accuracy are mentioned, but the corresponding results cannot be found in the table.
4. Besides the improvement in accuracy, are there any other improvements such as training time? I guess the training time will increase.
5. Are there any results about handling data with outliers?

---

### Official Review · Reviewer_jhZn · 2023-11-06

**Soundness:** 3 good
**Presentation:** 2 fair
**Contribution:** 2 fair
**Rating:** 5
**Confidence:** 2

**Summary:**

In this paper, the author first analyzes the advantages and disadvantages of discretization in numerical feature learning. To address the issue of inadequate training in deep learning with numerical inputs, the paper introduces a MultiHot Embedding method for the numerical value representation in deep learning.

**Strengths:**

1. The paper organization is clear, which makes it easy to read.
2. The advantages and disadvantages of discretization have been analyzed in detail.

**Weaknesses:**

1. Experiment section is weak, only two models are not enough to prove the claim. The ResNet and MLP structures are simple and easy to quantize and train. This cannot reflect the generality of the method proposed in this paper, so it's unclear if findings may generalize.
2. While the paper mentions reasons for not using the transformer model, it needs to provide a more detailed explanation, such as offering practical examples, data, or previous research examples.
3. The experimental analysis in this paper is not comprehensive enough. In Table 1, the method proposed in this paper did not demonstrate an advantage in some of the datasets, and the reasons for this phenomenon need to be explained.
4. This paper lacks ablation experiments, such as a comparison between MultiHot Embedding and one-hot Embedding.

**Questions:**

1. What are the results when only using ResNet or MLP in the experiment?
2. Will this method incur additional training costs? How practical is the application of this technology?

---

### Official Review · Reviewer_cyHj · 2023-11-06

**Soundness:** 3 good
**Presentation:** 3 good
**Contribution:** 2 fair
**Rating:** 3
**Confidence:** 4

**Summary:**

In this paper, the authors propose a MultiHot Embedding model for embedding numerical features. The main idea is to allow neighboring numerical features to leverage common embeddings to capture the context and retains the numerical relationship. Also, the authors show that such a MultiHot Embedding is much easier to optimize compared with the numerical feature embedding methods.

**Strengths:**

1. The idea of using MultiHot Embedding for embedding numerical features seems intuitive and useful.
2.  The paper is generally well organized and well-written.

**Weaknesses:**

1. The novelty of the proposed method is very limited. Basically, the proposed method allows numerically similar features to share similar embeddings. The idea has been leveraged in many time-series prediction tasks for embedding sentence via word embeddings.

2. There are several hyperparameters in the proposed method which makes the method hard to be applied in real-world problems. For example, the parameter m can be hard to decide given a new problem.

3. The paper lacks theoretical analysis of the proposed method so it is not clear if the mentioned properties are generally true.

**Questions:**

1. The authors use Time of Day as an example for embedding 8:59 and 9:00. If we assume we want to embed 11:59 and 12:00, although numerically they are close, but  they are actually very different in the context. How would the proposed MultiHot Embedding handle this case?

2. How to decide the parameter m?

---

### Official Review · Reviewer_AWvM · 2023-11-07

**Soundness:** 2 fair
**Presentation:** 2 fair
**Contribution:** 2 fair
**Rating:** 5
**Confidence:** 2

**Summary:**

The paper introduces a novel method for numerical feature representation in deep learning models, called MultiHot Embedding. The method discretizes numerical values into bins and activates multiple neighboring bins to form a MultiHot representation. The MultiHot representation preserves the numerical relationship of the values and enhances the training efficiency of the embedding. The paper evaluates the method on six public datasets with different neural network architectures and shows that it outperforms the baseline methods in most cases. The paper also analyzes the sensitivity of the method to different discretization configurations. The paper contributes to the field of feature representation learning, especially for numerical features in tabular data.

**Strengths:**

The author presents a novel and original method for numerical feature representation in deep learning models, called MultiHot Embedding. The method addresses the challenges of information loss and insufficient training caused by discretization of numerical features. The method is based on a simple but effective idea of activating multiple neighboring bins to form a MultiHot representation, which preserves the numerical relationship of the values and enhances the training efficiency of the embedding. And the author also provides a theoretical analysis of the method and its advantages over existing methods. The paper evaluates the method on six public datasets with different neural network architectures and shows that it outperforms the baseline methods in most cases. The paper also analyzes the sensitivity of the method to different discretization configurations and provides some insights into the optimal choices.

**Weaknesses:**

The paper does not provide a clear definition or formalization of the problem of numerical feature learning and the objective of the method. The paper only gives a brief introduction and motivation of the problem, but does not specify the input, output, and constraints of the method. The paper also does not explain how the method optimizes the embedding and the loss function of the neural network. A clear problem definition and formalization would help the readers to understand the method better and to compare it with other methods more easily.
The paper does not provide a thorough analysis of the effect of the discretization width and the number of activated bins on the performance of the method. The paper only reports the results of a single configuration for each dataset, without showing the sensitivity of the method to different choices of these parameters. The paper also does not provide any guidance or criteria for choosing the optimal configuration for a given dataset or task. A more comprehensive analysis of these parameters would help the readers to evaluate the robustness and generalizability of the method and to apply it to their own problems.

**Questions:**

1.How to define the optimal discretization width for a given numerical feature? Is there a general criterion or a data-dependent heuristic that you use to determine the best width?
2.How to interpret the learned embedding vectors of the MultiHot Embedding? Do they capture any semantic or structural information of the numerical features? Can you provide some visualization or analysis of the embedding space?

---

### Meta-Review · Area_Chair_mA7z · 2023-12-11

**Metareview:**

The paper proposes a method for learning multi-hot feature embeddings by discretizing the features into bins and activating multiple bins to form a multihot representation. Reviewers find the paper to be quite limited in scope and technical novelty, while also raising a question on the motivation of the paper since modern deep learning can handle numerical features well enough. Authors haven't responded to the reviews. Due to these limitations, the paper in its current form isn't suitable for publication at ICLR.

**Justification For Why Not Higher Score:**

Limited technical novelty and scope; questionable motivation behind the method which hasn't been responded to by the authors.

**Justification For Why Not Lower Score:**

N/A

---

### Decision · Program_Chairs · 2024-01-16

Reject